# Protecting the Entanglement of X-Type Systems via Weak Measurement and Reversal in the Generalized Amplitude Damping Channel

**DOI:** 10.3390/e27040350

**Published:** 2025-03-27

**Authors:** Meijiao Wang, Haojie Liu, Lianzhen Cao, Yang Yang, Xia Liu, Bing Sun, Jiaqiang Zhao

**Affiliations:** School of Physics and Electronic Information, Weifang University, Weifang 261061, China; wangmeijiao@wfu.edu.cn (M.W.);

**Keywords:** quantum entanglement, decoherence, weak measurement and reversal, generalized amplitude damping channel

## Abstract

The study of system evolution in generalized amplitude damping is of great significance in quantum information science and quantum computing. As an important quantum noise channel, the generalized amplitude damping channel can describe the general phenomenon of the energy dissipation effect in quantum systems at finite temperature. In this paper, we study the use of weak measurement and reversal to protect the entanglement of X-type systems in generalized amplitude damping channels, and give an experimental scheme. The results show that the closer to zero the temperature environment, the better the protection effect of weak measurement and reversal, but the lower the success rate. Therefore, when choosing an experimental environment, it is important to consider not only the temperature factor but also the probability of success. Because all quantum systems work at finite temperatures, it is hoped that the study of generalized amplitude damping channels can help design more robust quantum algorithms and protocols to improve the efficiency and stability of quantum information processing.

## 1. Introduction

Quantum entanglement is a valuable but fragile core resource in quantum information science. Due to environmental interaction [1,2], the entangled system is prone to decoherence, which leads to the loss of entangled information and may lead to the failure of quantum information processing schemes [3,4]. This presents a huge hurdle that must be overcome in the pursuit of practical quantum computing and quantum simulation. Among the myriad challenges in various applications, mitigating decoherence remains the most important. Therefore, maximizing the preservation of quantum entanglement during operation has been a major research goal. This paper focuses on the use of weak measurements as a means of maintaining quantum entanglement [5,6,7,8,9,10].

Researchers have conducted extensive studies on channels under zero-temperature conditions [11,12,13,14,15,16,17,18,19,20]. At zero temperature, they have successfully combined weak measurement with quantum measurement reversal (WMR), exploring an effective strategy to safeguard qubit systems against decoherence effects. Weak measurement (WM) enables the acquisition of crucial information about the system’s state without destroying the quantum state itself, thereby providing a solid foundation for subsequent interventions. Subsequently, the application of quantum measurement reversal (QMR) technology effectively reverses the minor disturbances introduced by WM to the system, significantly mitigating the occurrence of decoherence.

From an applied perspective, the study of amplitude damping (AD) channels at finite temperatures holds even greater significance for understanding the behavior of quantum systems in complex environments. This is because energy dissipation during interaction processes is not merely a theoretical concept but a ubiquitous phenomenon across numerous physical systems, including the spontaneous emission of photons by atoms, the evolution of spin systems under the influence of temperature, and the scattering and attenuation of photons in interferometers or cavities [21,22,23,24,25,26]. Similarly, in quantum computation and communication, qubits inevitably interact with their surrounding environment, leading to decoherence and the dissipation of quantum information [27,28,29,30]. By delving into the evolution of systems within generalized amplitude damping channels, we can devise more robust quantum algorithms and protocols, thereby enhancing the efficiency and stability of quantum information processing.

Drawing upon the aforementioned research findings in zero-temperature environments and the protection of Bell state entanglement at finite temperatures [31,32,33], this paper will investigate the evolution of quantum entanglement dynamics arising from the coupling of X-type systems in the generalized amplitude damping (GAD) channel. Furthermore, we will explore the protective effects of introducing WMs on the entanglement of these systems, aiming to provide a more realistic and comprehensive understanding of the decoherence dynamics of qubit systems. Ultimately, our work seeks to offer data support for laboratory simulations and constructions, as well as crucial theoretical foundations and practical guidance for the protection and control of quantum information.

The rest of this paper is structured as follows. The physical model and dynamics process of the system in the GAD channel is introduced in Section 2. In Section 3, the protection of X-type system with the pre-WM and post-QMR operation is discussed, as well as the probability of success. The conclusion is presented in Section 4.

## 2. Physical Model and Dynamics Process of the System in the GAD Channel

The GAD channel plays a key role in quantum information theory, providing a theoretical framework for describing the energy dissipation effects between quantum systems and their environments. This channel not only encompasses the traditional AD channel but also extends its reach to encompass the dynamic behavior of quantum systems under finite-temperature environments. The GAD channel is characterized by two main parameters that intricately characterize it: the environmental temperature parameter *r* and the dissipation parameter *d*:(1)$$\begin{array}{c} E_{0}=\sqrt{r}\left(\begin{array}{cc} 1 & 0\\ 0 & \sqrt{1-d} \end{array}\right),E_{1}=\sqrt{r}\left(\begin{array}{cc} 0 & \sqrt{d}\\ 0 & 0 \end{array}\right),\\ E_{2}=\sqrt{1-r}\left(\begin{array}{cc} \sqrt{1-d} & 0\\ 0 & 1 \end{array}\right),E_{3}=\sqrt{1-r}\left(\begin{array}{cc} 0 & 0\\ \sqrt{d} & 0 \end{array}\right). \end{array}$$ Here, r∈[0,1] signifies the influence of ambient temperature, while *d* is intimately tied to the dissipation process of the quantum system, specifically expressed as $d=1-e^{-t/T}$, where *t* represents time and *T* is a constant characterizing the dissipation rate. When *r* equals 0 or 1, the GAD channel reduces to the traditional AD channel, describing energy dissipation solely within a single environment.

We assume that the initial entangled state of the X-type is formulated as outlined below:(2)$$\rho_{0}=\frac{1}{4}\left(I⊗I+\sum_{i=0}^{3}c_{i}\sigma_{i}^{A}⊗\sigma_{i}^{B}\right).$$ Within the context of a two-qubit system, the identity operator is denoted by *I*, whereas $\sigma_{i}^{A}$ and $\sigma_{i}^{B}$ signify the Pauli operators that operate on the respective qubits held by Alice and Bob. The coefficients $|c_{i}|$, constrained by 0 ⩽ $|c_{i}|$ ⩽ 1, ensure the normalization and positivity of the density matrix $\rho_{0}$. To investigate the robustness of the initial X-type state against decoherence, we will explore three distinct cases stemming from this initial state. Specifically, we will analyze the Bell state with $|c_{1}\left|\;=\;\right|c_{2}\left|\;=\;\right|c_{3}|\;=\;1$, the Werner state where $|c_{1}\left|\;=\;\right|c_{2}\left|\;=\;\right|c_{3}|\;=\;0.8$, and finally, the general state where $|c_{1}|\;=\;0.7$, $|c_{2}|\;=\;0.9$, and $|c_{3}|\;=\;0.4$. These cases will allow us to assess the protection afforded to the initial X-type state under various conditions.

Figure 1a depicts the complex evolution of the X-type initial entangled states under the influence of GAD channels, marked by the physical tracks labeled ①. Using the power of linear optics, the GAD noise channel is realized on an experimental device by cleverly utilizing photon polarization as a qubit. Transitioning to Figure 1b, we use a type-I crystal to generate polarized entangled photon pairs, inspired by [29]. This innovative device provides a powerful platform for in-depth study and analysis of the complex effects of GAD noise on quantum entangled states.

As depicted in Figure 1b, the GAD channel comprises two distinct processes arising from finite-temperature thermal noise. One process involves the relaxation of the excited state, encapsulated by the operators $E_{0}$ and $E_{1}$. Notably, in the absence of any other mechanisms, the GAD channel reverts to the AD channel, a depiction of zero-temperature thermal noise, solely through this relaxation process. The AD channel, alongside the operators $E_{0}$ and $E_{1}$, have been previously documented in Refs. [4,5]. To physically realize $E_{0}$ and $E_{1}$ (as well as $E_{2}$ and $E_{3}$, their counterparts), a Sagnac-like interferometer (SLI) and an additional beam splitter (BS) are employed [4,5]. The SLI itself consists of a polarizing beam splitter (PBS), three mirrors, and two half-wave plates (HWPs). The dissipation parameter *d* of the channel can be precisely tuned by adjusting one of the HWPs while maintaining the other at 0 degrees. Specifically, for the implementation of $E_{0}$ and $E_{1}$, the HWP in the reflected path (associated with the vertical polarization |V〉) remains at 0 degrees, while the angle of the HWP in the transmitted path (horizontal polarization |H〉) is set according to the desired dissipation parameter *d*, as outlined in [19]. The two outputs from the SLI directly correspond to the realization of $E_{0}$ and $E_{1}$, which are then incoherently combined at the additional BS. To implement $E_{2}$ and $E_{3}$, the roles of |H〉 and |V〉 are simply swapped. The adjustable attenuators are utilized to set the channel dissipation parameter *d*, ensuring that the relative probabilities of the operations $E_{i}$ align with their theoretical values $d_{i}$. Ultimately, all paths converge at the final BS. Any random phases introduced by the SLI or other optical components are mitigated through the use of two quarter-wave plates + a half-wave plate (QHQ) setup, ensuring the integrity of the overall process.

It is presumed that both qubits are subjected to an independent yet identical GAD channel in Figure 1a. Consequently, the initial pure state inevitably transforms into a mixed state, which can be described by(3)$$\rho_{0}\to \rho =\sum_{j=0}^{15}K_{j}\rho_{0}K_{j}^{+},$$
where $K_{j}=E_{i}⊗E_{k}$(i,k=0,1,2,3;j=0,...,15) are the Kraus operators.

By bringing Equations (1) and (2) into Equation (Equation 3), a concrete expression can be obtained after the complex evolution process of the X-type entangled system, which is the entanglement evolution process given in Figure 1. Therefore, the density matrix of the X-type system can be expressed as(4)$$\begin{array}{cc} \rho & =\frac{1}{4}\left(\begin{array}{cccc} a & 0 & 0 & b\\ 0 & c & h & 0\\ 0 & h^{*} & e & 0\\ b^{*} & 0 & 0 & f \end{array}\right), \end{array}$$
with(5)$$\begin{array}{cc} a & =r^{2}\left[\left(1+c_{3}\right)\left(1+d^{2}\right)+2\left(1-c_{3}\right)d\right]+2r\left(1-r\right)\left[\left(1+c_{3}\right)\left(1-d\right)+\left(1-c_{3}\right)d\left(1-d\right)\right]+{\left(1-r\right)}^{2}\left(1+c_{3}\right){\left(1-d\right)}^{2},\\ b & =b^{*}=\left(c_{1}-c_{2}\right)\left(1-d\right),\\ c & =e=\left(2r^{2}-2r+1\right)\left[\left(1-c_{3}\right)\left(1-d\right)+\left(1+c_{3}\right)d\left(1-d\right)\right]+2r\left(1-r\right)\left[\left(1-c_{3}\right)\left(1-d+d^{2}\right)+2\left(1+c_{3}\right)d\right],\\ h & =h^{*}=\left(c_{1}+c_{2}\right)\left(1-d\right),\\ f & =r^{2}\left(1+c_{3}\right){\left(1-d\right)}^{2}+2r\left(1-r\right)\left[\left(1+c_{3}\right)\left(1-d\right)+\left(1-c_{3}\right)d\left(1-d\right)\right]+{\left(1-r\right)}^{2}\left[\left(1+c_{3}\right)\left(1+d^{2}\right)+2\left(1-c_{3}\right)d\right]. \end{array}$$

The entanglement between the qubits can be quantitatively assessed using the measure of concurrence, as outlined in [34]. Specifically, the concurrence of the density matrix ρ is denoted as(6)$$C\left(\rho \right)=max\{0,\Lambda_{1}\equiv 2(|b|-\sqrt{ce}),\Lambda_{2}\equiv 2(|h|-\sqrt{af})\}.$$ In scenarios where the parameter $\Lambda_{1}$ or $\Lambda_{2}$ exceeds zero, the concurrence takes the value of $\Lambda_{1}$ or $\Lambda_{2}$, signifying the presence of entanglement. Conversely, if $\Lambda_{1}$ and $\Lambda_{2}$ are non-positive, the concurrence assumes a value of zero, indicating the absence of the entanglement.

By combining Equations (4)–(6), one can easily calculate the concurrence of the system after evolution. In Figure 2, the concurrence of the X-type initial state is plotted as a function of the dissipation parameter *d* with different environmental temperature parameters *r*, considering various initial states including the Bell state, Werner state, and general state. As can be seen from Figure 1, for the three initial states, when the initial dissipation parameter *d* is small, the difference in the entanglement under different temperature parameters *r* is minimal. As the dissipation parameter *d* increases, the difference in the entanglement under different temperature parameters *r* also becomes more significant, and entanglement sudden death even occurs in the system under higher temperature parameters *r*. Given a fixed dissipation parameter *d*, Figure 2 also reveals that as the temperature parameter *r* increases, the entanglement of the system increases. This is because although the temperature parameter *r* is increasing, it can be seen from Equation (Equation 1) that the larger *r* is, the lower the corresponding environment temperature is, and more states are in the ground state. When *r* is low and the corresponding temperature is high, the system will absorb energy from the environment and transition to the excited state. Compared with the ground state, the excited state is more likely to interact with the environment, thus reducing the entanglement of the system. Therefore, when selecting the experimental environment, it is advisable to choose an environment with as low a temperature as possible to reduce the loss of the entanglement. Additionally, we can also utilize the WMR scheme to protect the entanglement of the system.

## 3. The Protection of X-Type System

### 3.1. Weak Measurement

WM is a measurement technique in quantum mechanics used to obtain system information without causing significant disturbance. Unlike traditional strong measurements, WM interacts weakly with the system, extracting only a small fraction of quantum information, thereby allowing the system to remain in an approximately undisturbed state. This mode of measurement is particularly crucial in quantum information processing and quantum computing, as it enables the necessary monitoring and control of quantum states while protecting them from decoherence and entanglement loss.

For qubits, WM can be expressed as a non-unitary quantum operation, which can be expressed by the density matrix as follows:(7)$$M_{wm}=\left(\begin{array}{cc} 1 & 0\\ 0 & \sqrt{1-p} \end{array}\right),$$
where 0≤p≤1 represents the strengths of WM for transitions from the state |1〉 to |0〉 in the qubit system. Since the computational state |0〉 does not interact with the environment, the system’s resistance to decoherence is enhanced after the WM operator $M_{WM}=M_{wm}⨂M_{wm}$ is performed, where $M_{wm}$ denotes the WM operator acting on the individual qubit. By using the same interferometer configuration, we are also able to implement WMs, the only difference being that we only collect photons from the output path 0, as shown in Figure 1b. It is worth noting that the dotted green box in Figure 1b represents the WM operation we performed within the framework of the experiment.

### 3.2. Quantum Measurement Reversal

After performing the WM, the initial state can be effectively restored by designing appropriate QMR operations. Like WM, the QMR operator is also a non-unitary quantum operation, and the density matrix can be formulated as(8)$$M_{qmr}=\left(\begin{array}{cc} \sqrt{1-q} & 0\\ 0 & 1 \end{array}\right),$$
where q∈[0,1] is the strength of QMR, demonstrating the intricate interplay between measurement and its reversal in the realm of quantum mechanics, and $M_{QMR}=M_{qmr}⨂M_{qmr}$. The QMR process can be systematically constructed by sequentially applying three distinct operations to each two-level quantum system: firstly, a bit-flip operation *F*, followed by a WM, and finally, another bit-flip operation *F*:(9)$$M_{qmr}=FM_{wm}F,$$
where the bit-flip operation *F* is represented by a matrix(10)$$F=\left(\begin{array}{cc} 0 & 1\\ 1 & 0 \end{array}\right).$$

### 3.3. The Evolution of the System After Pre-WM and Post-QMR Operations

If the system passes directly through the GAD channel, the entanglement will decay rapidly. To mitigate this effect, we implemented a pre-WM operation before the system passed through the GAD channel, as shown in Figure 1a (specifically, the dotted path labeled with ②), with the linear optics experiment also shown in Figure 1b. Subsequently, the system evolved over a period of time, and we finally implemented post-QMR operations. The final evolution result of the system is(11)$$\rho_{WMR}=M_{QMR}\varepsilon [M_{WM}\rho_{0}M_{WM}^{+}]M_{QMR}^{+}.$$

By combining Equations (1) and (7)–(9), the density matrix of the system can be obtained as(12)$$\begin{array}{cc} \rho_{WMR} & =\frac{1}{4P}\left(\begin{array}{cccc} a^{\prime }{\left(1-q\right)}^{2} & 0 & 0 & b^{\prime }\left(1-q\right)\\ 0 & c^{\prime }\left(1-q\right) & h^{\prime }\left(1-q\right) & 0\\ 0 & h^{{*}^{\prime }}\left(1-q\right) & e^{\prime }\left(1-q\right) & 0\\ b^{{*}^{\prime }}\left(1-q\right) & 0 & 0 & f^{\prime } \end{array}\right), \end{array}$$
with(13)$$\begin{array}{cc} a^{\prime } & =r^{2}\left\{\left(1+c_{3}\right)\left[1+d^{2}{\left(1-p\right)}^{2}\right]+2\left(1-c_{3}\right)d\left(1-p\right)\right\}+{\left(1-r\right)}^{2}\left(1+c_{3}\right){\left(1-d\right)}^{2}\\ \; & +2r\left(1-r\right)[\left(1+c_{3}\right)\left(1-d\right)+\left(1-c_{3}\right)d\left(1-d\right)\left(1-p\right)],\\ b^{\prime } & =b^{{*}^{\prime }}=\left(c_{1}-c_{2}\right)\left(1-d\right)\left(1-p\right),\\ c^{\prime } & =e^{\prime }=r^{2}\left[\left(1-c_{3}\right)\left(1-d\right)\left(1-p\right)+\left(1+c_{3}\right)d\left(1-d\right){\left(1-p\right)}^{2}\right]\\ \; & +{\left(1-r\right)}^{2}\left[\left(1-c_{3}\right)\left(1-d\right)\left(1-p\right)+\left(1+c_{3}\right)d\left(1-d\right)\right]\\ \; & +2r\left(1-r\right)[\left(1-c_{3}\right)\left(1-d+d^{2}\right)\left(1-p\right)+\left(1+c_{3}\right)d+\left(1+c_{3}\right)d{\left(1-p\right)}^{2}],\\ h^{\prime } & =h^{{*}^{\prime }}=\left(c_{1}+c_{2}\right)\left(1-d\right)\left(1-p\right),\\ f^{\prime } & =r^{2}\left(1+c_{3}\right){\left(1-d\right)}^{2}{\left(1-p\right)}^{2}\\ \; & +{\left(1-r\right)}^{2}\left[\left(1+c_{3}\right)d^{2}+\left(1+c_{3}\right){\left(1-p\right)}^{2}+2\left(1-c_{3}\right)d\left(1-p\right)\right]\\ \; & +2r\left(1-r\right)[\left(1+c_{3}\right)\left(1-d\right){\left(1-p\right)}^{2}+\left(1-c_{3}\right)d\left(1-d\right)\left(1-p\right)],\\ P & =\frac{1}{4}\left[a^{\prime }{\left(1-q\right)}^{2}+c^{\prime }\left(1-q\right)+e^{\prime }\left(1-q\right)+f^{\prime }\right]. \end{array}$$ Here, *P* stands for the success probability.

By substituting Equations (12) and (13) into (6), the entanglement of the system with the addition of WMR can also be easily calculated. If under the same temperature and damping parameters, after the implementation of the weak measurement scheme, the concurrence is larger than that without implementation, it is considered that the weak measurement scheme can improve the entanglement of the system, that is, it has played a protective effect. In order to maximize the recovery of the entanglement, we cleverly borrow the mathematical formula $a^{2}+b^{2}⩾2ab$ from Appendix E of Ref. [32], and we fix the weak measurement strength *p* and maximize concurrence by varying the measurement reversal strength *q*. Since the measurement reversal strength *q* appears only in the denominator, P, we turn to minimize the success probability *P* by adjusting the measurement reversal strength *q*, which establishes the relationship between the weak measurement strength *p* and the measurement reversal strength *q*, and optimize the entanglement of the system. Based on the theoretical calculation results, Figure 3 shows the concurrence for three initial states as the function of the WM strength *p* and the dissipation parameter *d* with different temperature parameters *r* after the WMR operation. In Figure 3, the entanglement corresponding to a weak measurement strength *p* equalling zero is the entanglement of the system without the implementation of the WMR scheme. It can be seen from the longitudinal comparison in Figure 3 that the three initial states have the same variation trend with the WM strength *p* and the dissipation parameter *d*, and there are only differences in value. The horizontal comparison shows that with the increase in the temperature parameter *r*, the protective effect of WMR becomes weaker. If the temperature continues to rise, the entanglement will decrease with the increase in the weak measurement strength *p*, as shown in Figure 4, so that the weak measurement scheme will not play a protective role, and even WMR will have the opposite effect. Therefore, when operating the experiment, a low-temperature environment is selected as far as possible to maximize the protection effect of WMR to maintain the entanglement of the system.

In order to show the protection effect of WMR more clearly, a comparison image of the three initial states with or without WMR protection at lower temperatures is given in Figure 5. It can also be seen from Figure 5 that the closer the environment is to zero temperature, the better the protection effect of WMR. In the ideal zero-temperature environment in Figure 5a, the WMR will restore the system’s entanglement to its initial value, but the success probability *P* for this result is slim. It can be seen that the environment has a great influence on the system entanglement. It is necessary to approach the ideal experimental environment.

### 3.4. The Probability of Success for the System with the WMR Operation

Although the protection effect of the entanglement is very good at low temperatures, the WM is a local collapse measurement based on von Neumann measurement and positive-operator value measurement, and it is a non-unitary quantum operation, so it is accompanied by a certain probability. Figure 6 and Figure 7 show the success probabilities of the three initial states at different temperature parameters *r*. It can be seen from Figure 6 that the evolution trend of the success probability *P* of the three initial states is almost the same, and the greater the WM strength *p* and dissipation parameter *d*, the lower the success probability. Since the change trend of the success probability *P* of the three states is similar, only the images of the success probability *P* of the Bell state under different temperature parameters *r* are given in Figure 7. As the temperature parameter *r* decreases, the success probability *P* increases. Although the success probability *P* is increased, the probability of success *P* is exactly opposite to the protection effect of WMR, and the protection effect of WMR against the entanglement at high temperatures is indeed poor. Therefore, in order to maintain the entanglement of the system to the greatest extent, we should not only select the appropriate temperature, but also consider the success probability of the scheme.

### 3.5. Protecting Entanglement in Zero Temperature

It can be seen from the previous study that the protection effect of WMR in a zero-temperature environment is the best. This section gives an image of the evolution of entanglement with WM strength *p* and the dissipation parameter *d* in an ideal zero-temperature environment r=1. As can be seen from Figure 8, even if the dissipation parameter *d* is large, the entanglement of the system increases with the increase in the WM strength *p*, and when the WM strength *p* approaches 1, the entanglement returns to the initial value, but the success probability at this time also approaches 0.

## 4. Conclusions

In generalized amplitude damping channels, the evolution of quantum systems is not only affected by the internal dissipation mechanism, but also by the external temperature. This complex action makes the evolution of quantum states more complex and variable, and poses a higher challenge to the transmission and storage of quantum information. Therefore, the protection effect of weak measurement in generalized amplitude damping channels on X-type systems is studied in this paper. It is found that the scheme combining pre-weak measurement and post-quantum measurement reversal can effectively suppress decoherence in generalized amplitude damping channels. The lower the temperature, the better the protection effect of weak measurement on the system, but at the cost of a lower probability of success. At the same time, the entangled state of the system in the ideal zero-temperature environment is given. The entanglement of the system can be restored to the initial value by adjusting the weak measurement parameters, but the probability of success is 0.

Since all quantum systems operate at finite temperatures, this research has potential applications in practical quantum tasks. Based on the linear optical system, we also give the experimental diagram, hoping that this research can be realized in more quantum systems, which will help promote the rapid development of quantum computing, quantum communication, quantum sensing, and other technologies.

## Figures and Tables

**Figure 1 entropy-27-00350-f001:**
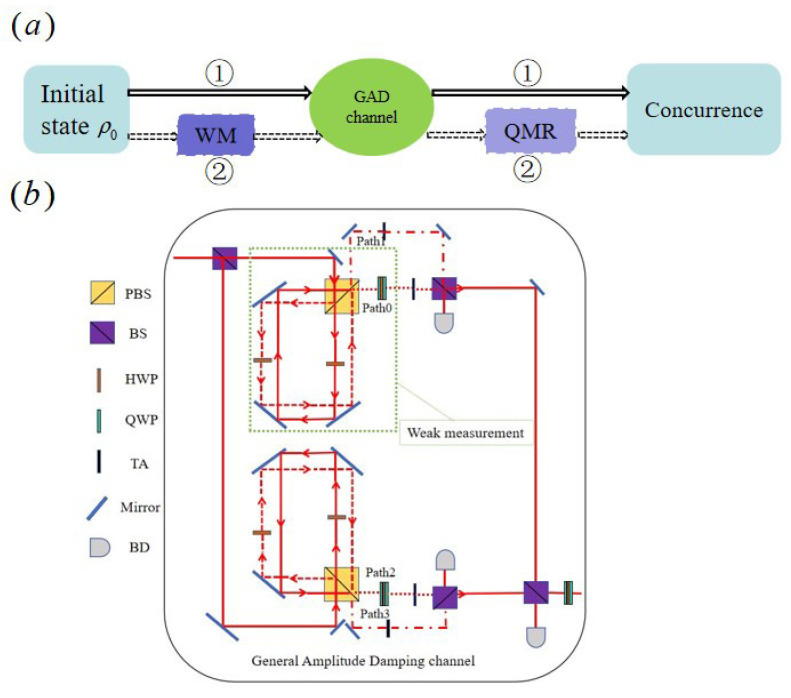
(**a**) The schematic diagram of the X-type initial state without or with the weak measurement and measurement reversal operation (the virtual box). (**b**) The experimental diagram of the GAD noise channel.

**Figure 2 entropy-27-00350-f002:**
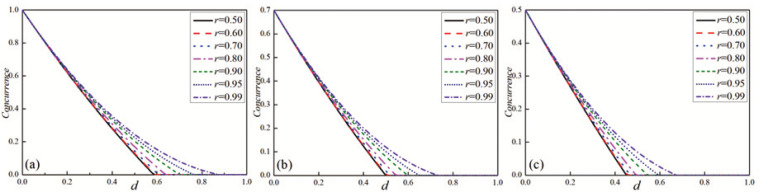
The concurrence for three initial states as the function of the dissipation parameter *d* with different temperature parameters *r*. (**a**) Bell state, (**b**) Werner state, and (**c**) general state.

**Figure 3 entropy-27-00350-f003:**
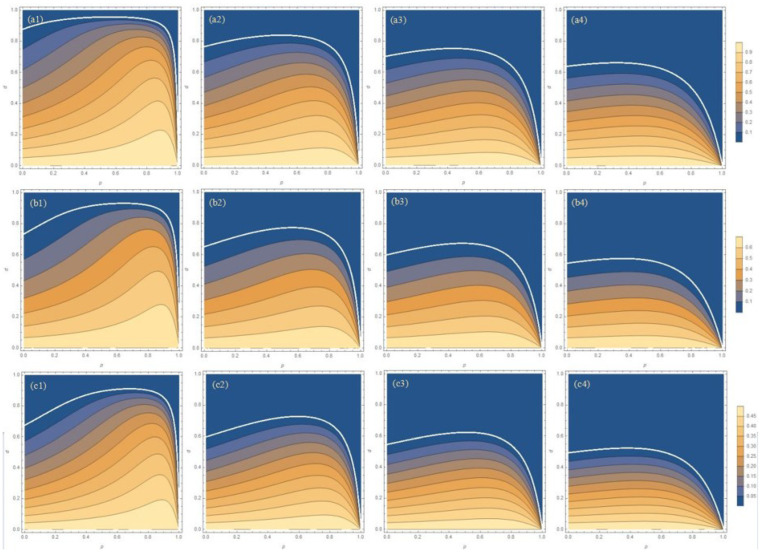
The concurrence for three initial states as the function of WM strength *p* and the dissipation parameter *d* with different temperature parameters *r*. (**a1**–**a4**) Bell state, (**b1**–**b4**) Werner state, and (**c1**–**c4**) general state, i=1,2,3,4; (*L*1) r=0.99, (*L*2) r=0.95, (*L*3) r=0.90, and (*L*4) r=0.80, L=a,b,c.

**Figure 4 entropy-27-00350-f004:**
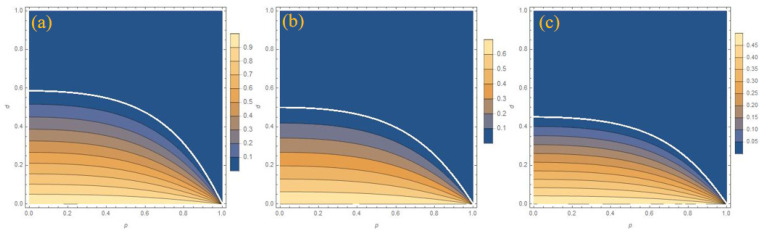
The concurrence for three initial states as the function of WM strength *p* and the dissipation parameter *d* with the temperature parameter r=0.5. (**a**) Bell state, (**b**) Werner state, and (**c**) general state.

**Figure 5 entropy-27-00350-f005:**
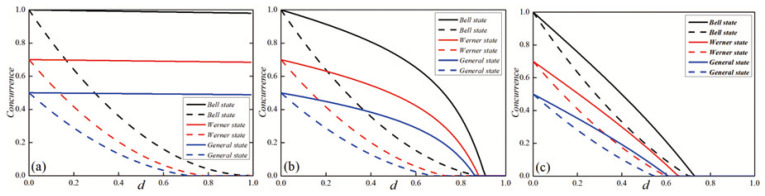
The comparison of the concurrence for three initial states with or without protection at different temperature parameters *r*. (**a**) r = 1, p = 0.99, (**b**) r = 0.99, p = 0.89, and (**c**) r = 0.9, p = 0.65. The solid (dashed) lines are results with (no) WMR protection. Different colors represent different initial states.

**Figure 6 entropy-27-00350-f006:**
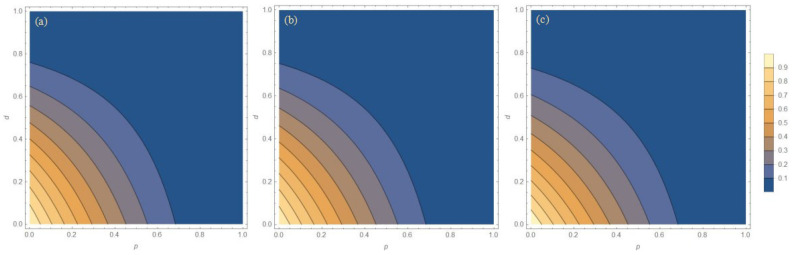
The success probability *P* for three initial states as the function of WM strength *p* and the dissipation parameter *d* with temperature parameter r = 0.99. (**a**) Bell state, (**b**) Werner state, and (**c**) general state.

**Figure 7 entropy-27-00350-f007:**
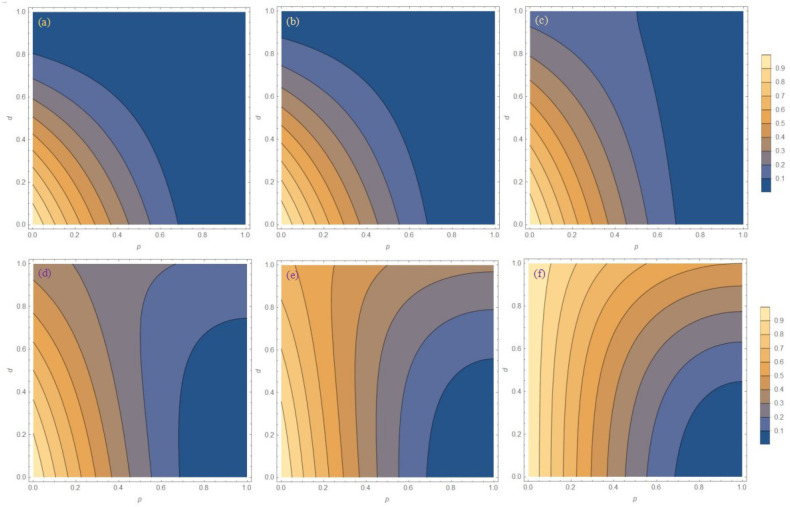
The success probability *P* for the Bell state as the function of WM strength *p* and the dissipation parameter *d* with different temperature parameters *r*. (**a**) r = 0.95, (**b**) r = 0.90, (**c**) r = 0.80, (**d**) r = 0.70, (**e**) r = 0.60, and (**f**) r = 0.50.

**Figure 8 entropy-27-00350-f008:**
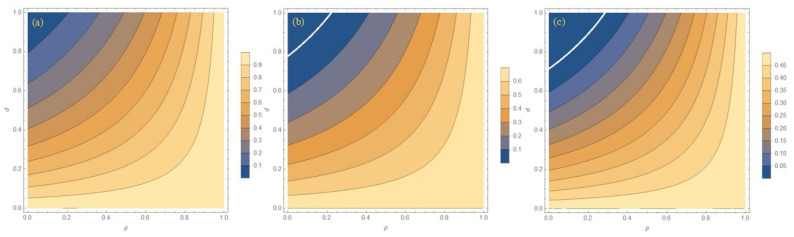
The concurrence for three initial states as the function of WM strength *p* and the dissipation parameter *d* with r = 1. (**a**) Bell state, (**b**) Werner state, and (**c**) general state.

## Data Availability

Data is contained within the article.

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
