# Peer review of "Protecting the Entanglement of X-Type Systems via Weak Measurement and Reversal in the Generalized Amplitude Damping Channel"

_entropy, 2025, doi:10.3390/e27040350_

Round 1

Reviewer 1 Report

Comments and Suggestions for Authors

In this manuscript, the authors investigated the scheme to protect the entanglement of X-type state using the weak measurement and reversal in the generalized amplitude damping channels. Indeed, a lot of similar works have been published in the past several years, some conclusions of this paper are predictable. In the present study, no further new results can be found. Therefore, in my opinion, this work does not meet the criteria of Entropy.

Comments on the Quality of English Language

The English could be improved to more clearly express the research

Reviewer 2 Report

Comments and Suggestions for Authors

In the manuscript titled "Protecting the Entanglement of X-Type Systems via Weak Measurement and Reversal in the Generalized Amplitude Damping Channel," the authors study the use of weak measurement and its reversal to protect entanglement in X-type systems within generalized amplitude damping channels. Through their analysis, they demonstrate that the combination of a pre-weak measurement and a post-quantum measurement reversal is effective in counteracting decoherence in such channels. Additionally, the authors propose an experimental scheme to support their approach. Their results indicate that the closer the environmental temperature is to zero, the better the protection effect of weak measurement; however, this improvement comes at the cost of a lower success probability.

The manuscript presents interesting results, displays clear written communication and demonstrates an appropriate structure. The results are robust and presented in an appropriate manner. It is recommended that this manuscript be considered for publication in Entropy, provided that the following points are addressed or taken into consideration in the revised version:

1. Equation 1 with the Kraus operators of the GAD channel has a formatting issue and appears to be cut off. 

2. The word "comprehensive/comprehensively" appears three times within five lines (lines 123–128). Please consider replacing some occurrences with synonyms. 

3. Do the conditions b = b* and d = d* still represent a generic X-state? 

4. The symbol d is used both as a matrix element in the density matrix (Eq. 4) and as the dissipation parameter of the GAD channel (Eq. 1). Please choose a different letter for one of these to avoid confusion. 

5. Is Eq. 6 correct in expressing concurrence as the maximum of three values? Does this hold for this specific class of states? In general, concurrence is given by the maximum between zero and the difference of the square roots of the ordered eigenvalues of \rho \tilde{\rho}. 

6. What does the uppercase P represent in the last line of Eq. 13? Is it the success probability? Please clarify. 

7. While the curves in the figures are generally of good quality, the legends are difficult to read. The color maps have very small scale bars, and the axis labels and titles are too small and lack good resolution. In Figure 4, the legend inside the plot is unreadable. I recommend improving the figures' readability by increasing the font size of labels, legends, and axes.

8. Finally, how feasible do the authors believe the practical implementation of the proposed scheme is? Could they provide an evaluation of the difficulty of implementation, efficiency, or the gain in entanglement persistence?

Reviewer 3 Report

Comments and Suggestions for Authors

In this manuscript, the authors study the combined effects of the generalized amplitude damping (GAD) channel and weak measurement & reversal (WMR) protection on entanglement in a two-qubit system. The key findings can be summarized as follows:

  • At lower temperatures, the GAD channel has a weaker effect in destroying entanglement.
  • At lower temperatures, the weak measurement has a stronger effect in protecting entanglement.
  • While lower temperatures seem preferable for preserving entanglement, this is not the case. A lower temperature also reduces the success rate of weak measurement. Therefore, the choice for the environment temperature can be a tricky question (a good choice of the temperature is actually not given by the authors).

Comments on the novelty: This work is an extension of an earlier work that was done in 2014 (Phys Rev A 89, 022318). That earlier work studies pure Bell states as initial states, while the current work extends its initial state to a general form of X-matrix states.

Comments on the presentation: The language seems to be over-polished.

Regarding the scientific clarity of the manuscript, I have the following comments:

  • For the parameters c1,c2,c3 in the initial state, the authors’ claims seem to be inaccurate. When c1=c2=c3=1.0, the state is not a GHZ state. When c1=c2=c3=0.8, the state is not a Werner state. A clearer explanation seems to be needed. Also, the state Eq.(2) is not normalized.
  • Notations repeated: the letter “d” is used for both Eq.(1) and Eq.(4), but they represent different physical notions.
  • (7) and Eq.(8) specify the strength of weak measurements (p) and its reverse (q), but it is nowhere clarified how p is related to q.
  • In line 202, it is mentioned that “even if the temperature continues to rise, the WMR will have the opposite effect.” But this “opposite effect” is nowhere shown or depicted in the manuscript.
  • The “protection effect” by WMR is not defined. In the original literature, the strength of WMR is optimized to achieve maximal entanglement recovery, but this is not presented in the current work.
  • The concept of “success probability” in section 3.4 is not defined.
  • In line 149, the statement “when the excited states interact with the environment, it leads to a reduction in the entanglement” is problematic for two reasons. First, the sentence itself does not convey a precise meaning. Second, it fails to explain why higher temperatures result in lower entanglement values, which is its intended purpose in this context.

Based on the assessment of the paper’s novelty, presentation, and scientific clarity, it is difficult to determine the correctness of its results before the above points are addressed. Therefore, I recommend at least a major revision before considering its publication.

Comments on the Quality of English Language

Comments on the presentation: The language seems to be over-polished.

Round 2

Reviewer 1 Report

Comments and Suggestions for Authors

I appreciate the authors' efforts in addressing the reviewers' queries. However, I believe the manuscript still lacks sufficient novelty to be accepted in this journal. 

Comments on the Quality of English Language

 The English could be improved to more clearly express the research.

Author Response

Dear reviewer,

  Thank you very much for reviewing this study again. In this manuscript, we mainly study the evolution of the entanglement in the finite temperature amplitude damping channel and its protection by weak measurement scheme. We are sorry that we failed to meet your review requirements. We will continue to combine other existing schemes (for example, dynamic decoupling, error-correcting code) in the future to see if we can further improve entanglement or other more interesting results.

  Thanks again!

  Yours sincerely,

              Mei-Jiao Wang (for all the authors)

Reviewer 2 Report

Comments and Suggestions for Authors

The authors have satisfactorily addressed all my questions, and the article is accepted for publication. In my opinion, the figure axes and the color scales of the maps are quite difficult to read, but I leave this to the journal's editorial process.

Author Response

Dear reviewer,

  Thank you very much for your recognition of this study. For image issues, it may be tight to some extent due to tool or format issues, but if there is any doubt in the post editing, we are more than happy to continue to improve!

  Thanks again!

                                             Yours sincerely,

Mei-Jiao Wang (for all the authors)

Reviewer 3 Report

Comments and Suggestions for Authors

In this revised manuscript, the authors have made efforts to improve its quality. However, after careful considerations, I find that the current manuscript remains far from self-contained. My judgement does not concern the correctness of the equations derived within the text. Rather, my concern is that the presentation assumes prior familiarity with previous literature and omits many necessary explanations, making it difficult for readers to follow the ideas and the successive conclusions following within the text. I will elaborate on this point in the following paragraphs.

Comment on Response 1: In the response, the authors state that “the closer r is to 1, the better the result.” However, this appears to contradict their statement in the manuscript: “As r decreases, the success probability P increases… we should not only select the appropriate temperature but also consider the success probability of the scheme.” This apparent inconsistency raises concerns about the coherence of the main conclusion for the work.

Comment on Response 4: I now understand the reason for my previous confusion. This is due to an inconsistency within the text. The text explicitly states that ci>=0; however, achieving the desired Bell state and Werner state requires at least some of these parameters to be negative.

Comment on Response 6: In their response, the authors clarify how the relationship between p and q is derived. However, for reproducibility of the results, these explanations should be included in the manuscript itself. Specifically, the authors should have at least said something along the lines of: “We fix p and maximize concurrence by varying q. Since q appears only in the denominator, P, we turn to minimize P by adjusting q, which establishes the relationship between p and q.” Without this clarification, the readers can by no means understand what the authors are really doing, and would not be able to understand what it means by “we can obtain the relationship between p and q, and optimize the entanglement of the system.”

Comment on Response 7: The authors state that “If the temperature continues to rise, the entanglement will decrease with the increase of the weak measurement strength p.” However, this claim does not appear to be supported by any specific data or figures in the manuscript. It would be helpful for the authors to include a figure or additional evidence to support this conclusion.

Comment on Response 8: To explain the “protection effect,” the authors use phrases such as “Under the same parameters... it is considered that the protection effect has been achieved.” However, similar explanations should be included in the main text, as the term “protection effect” is introduced by the authors and requires further clarification. The readers do not have access to this private conversation.

Additionally, in the original work [32], concurrence was maximized by varying both p and q (though they used the notations m and n). In this work, however, the maximization is performed only on q. This is why the relationship between p and q can be derived, allowing the authors to draw plots with respect to parameter p. Including a second optimization over p seems reasonable to me personally, as the current introduction of the “protection effect” only focuses on entanglement protection “with” and “without” the implementation of WMR.

In summary: Given the limited novelty of the results presented in the text, as well as the fact that the story is far from self-contained, I tend to argue that the current manuscript reads more like a long appendix to prior literature rather than an independent research article. For this reason, I cannot recommend it for publication.

Comments on the Quality of English Language

Big improvement can be observed!

Round 3

Reviewer 1 Report

Comments and Suggestions for Authors

I appreciate the authors' efforts in addressing the reviewers' queries point by point.  Also, rich theoretical results are shown in the manuscript, while little physics behind the calculation has been analyzed. In addition, the authors may try to consider the case for general initial state besides X state. I will recommend the manuscript to be published after these questions are well discussed.

Comments on the Quality of English Language

The English could be improved to more clearly express the research.

Reviewer 3 Report

Comments and Suggestions for Authors

I appreciate the authors' efforts. My previous concerns have been fully addressed, and I think the current version is ready for publication in Entropy.

Author Response

Dear reviewer,

  Thank you very much for your recognition of this study. In the whole process of revising the manuscript, we felt your academic foundation and your rigorous research ideas, which benefited us a lot. The process of this revision is of great help to the writing ideas and language expression of other manuscripts in our later period.

  We all the authors thank you!

                                                                                      Yours sincerely,

Mei-Jiao Wang (for all the authors)